# The Regulating Effect of Urban Large Planar Water Bodies on Residential Heat Islands: A Case Study of Meijiang Lake in Tianjin

**Liuying Wang, Gaoyuan Wang, Tian Chen * and Junnan Liu** 

School of Architecture, Tianjin University, 92 Weijin Road, Nankai District, Tianjin 300072, China; wangliuying_1994@tju.edu.cn (L.W.); 1020206035@tju.edu.cn (G.W.); liujunnan_430@tju.edu.cn (J.L.)
* Correspondence: teec@tju.edu.cn

**Abstract:** Efficiently harnessing the urban cool island effect associated with large urban aquatic bodies holds significant importance in mitigating the urban heat island (UHI) effect and enhancing the quality of residential living. This study focuses on Tianjin's Meijiang Lake and its surrounding 47 residential areas, combining Landsat 8 remote sensing satellite data with geographic information system (GIS) buffer analyses and multiple linear regression analyses to reveal the summer thermal characteristics of residential waterfront areas with diverse spatial layouts. The results indicate that: (1) Meijiang Lake's effective cooling radius extends up to 130 m from the water's edge, achieving a maximum temperature reduction of 14.44%. Beyond 810 m, the cooling effect diminishes significantly. (2) Waterfront distance (*WD*), building density (*BD*), building width (*L*) and normalized difference vegetation index (*NDVI*) emerge as the primary factors influencing changes in average land surface temperature (Δ*LST*) in residential areas. The degrees of influence are ordered as follows: *BD* > *WD* > *NDVI* > *L*. "Dispersed" pattern residential areas exhibit the most favorable thermal environments, which are primarily influenced by *WD*, while "parallel" pattern residential areas demonstrate the least favorable conditions, primarily due to *WD* and *NDVI*. (3) The direct adjacency of residential areas to large-scale aquatic bodies proves to be the most effective approach for temperature reduction, resulting in a 5.03% lower average temperature compared to non-adjacent areas. Consequently, this study derives strategies for improving the thermal environment via the regulation of spatial planning elements in residential areas, including waterfront patterns, vegetation coverage, *WD*, and *BD*.

**Keywords:** urban heat island; thermal environment; residential waterfront area; spatial morphology; cold regions

## 1. Introduction

Alleviating the urban heat island effect is considered an essential measure for addressing climate change. Adverse urban heat environments can alter the wind patterns and water circulation processes between urban and rural areas [1], disrupt local energy balance, change the patterns of pollutant dispersion and diffusion, and have a significant impact on the comfort of living environments. Tianjin, located in the cold northern region of the North China Plain, is a major city in China with dense residential areas and a prominent urban heat island effect [2]. Surface urban heat (SUH) is sensitive to the spatiotemporal differentiation of urban canopy heat islands [3] and has advantages in studying surface characteristics and human activities. It is often used as a parameter to characterize the thermal environment features at multiple scales in urban areas. Scholars use remote sensing monitoring data to retrieve surface temperatures as a parameter for characterizing thermal environment features at multiple scales in cities, and the extensive coverage of satellite remote sensing data provides a data foundation for studying the distribution characteristics of the urban thermal environment [4,5]. Currently, research on the urban thermal environment mainly includes studies on the spatiotemporal characteristics of the thermal

environment and research on driving forces and mechanisms. Urban morphology, social driving forces, meteorological conditions, and physiological and biochemical conditions are all factors that influence changes in the urban thermal environment [6,7]. In terms of urban morphology, the factors influencing the thermal environment mainly consist of urban geometric structure [8], surface materials of land and buildings [9], urban ventilation [10], and vegetation and water bodies [11]. Multiple studies have concluded that urban geometric structure, vegetation, and water bodies are the primary factors causing variations in the urban summer thermal environment [12].

Water bodies, as blue patches within cities, exhibit a cool island effect during the summer due to their lower surface radiation and the cooling of surrounding air [13]. Apart from having a higher heat capacity [14] and promoting heat transfer via convective effects from evaporation [15], in regions with generally low humidity, water bodies increase local humidity, aiding heat exchange during the summer [16]. Additionally, in urban development, water bodies, as open spaces, can create wind corridors that facilitate airflow and promote urban cooling [17]. Tianjin, situated in the downstream area of the Haihe River Basin, is where five major tributaries of the Haihe River system converge. It is a city in the North China region and has a relatively high proportion of surface water, making the relationship between urban residential environments and surface water environments complex [18]. Tianjin's urban development has been closely linked to its waterways since its inception, and the relationship between residential areas and water bodies is inseparable [19]. Utilizing water bodies within the city to create a favorable microclimate for residential areas and harmonizing the relationship between water and urban residential environments are key concerns in the planning and development of Tianjin [20]. Urban water bodies can be divided into various types according to their morphological differences, among which planar water refers to the types of depressions formed by natural surfaces, such as lakes and reservoirs, while the opposite concept comprises linear water bodies dominated by rivers. Efficiently harnessing the cooling effects of large planar water bodies can not only create pleasant outdoor spaces and enhance the quality of living but also reduce heat-related health risks [21] and promote carbon emission reduction [22]. A favorable summer outdoor thermal environment can effectively reduce buildings' cooling loads [23,24] and decrease the use of appliances such as air conditioning. In research on strategies that improve urban thermal environments, the conclusions regarding the interaction between water bodies and urban spaces hold direct reference value.

Research on the thermal environment of water bodies in urban areas primarily focuses on two aspects:

(1)　The study of the cooling effect mechanism of water bodies in cities based on their own characteristics, such as radiation, evaporation, humidity changes, and the spatial and temporal characteristics of their size and morphology [25–27].

(2)　The mechanism of thermal environment changes with respect to the spatial characteristics between water bodies and the surrounding urban areas. Currently, most research on the impact mechanism of water bodies on the thermal effects of urban spaces is conducted at the macroscale, including studies on urban clusters [28] and city scales [29].

The primary research method involves the analysis of remote sensing satellite data, treating water bodies as part of the landscape pattern and studying their impact mechanisms on the thermal environment [30,31]. At the meso and micro levels, methods involve ground data measurement and computer simulations, mainly studying the impact mechanisms of water bodies on open spaces, such as surrounding streets [32], and vegetation configuration [33] in the thermal environment. Residential areas, as important components that bear the functions of cities and waterfront regions, have existing research results indicating that the thermal environment of residential areas is influenced by factors such as sky visibility [34] and building layout [35]. Moreover, Yang et al. [36] observed that green space ratio, building height, and building density have interactive effects on the thermal environment of residential areas. Song et al. [37], carrying out CFD (computational fluid

dynamics) simulations of wind and thermal environments in northern Chinese residential areas, observed that areas with low plot ratios and large ventilation corridors are more susceptible to the influence of water bodies. Research on the thermal environment of residential areas in waterfront regions is still limited, with some research results focusing on severely cold regions in China [38], regions with hot summers and cold winters [39], and regions with hot summers and mild winters [40]. There are significant variations in different climatic zones, and research on the thermal environment characteristics of residential waterfront areas in cold regions is still insufficient.

A review of studies shows that there is insufficient research on the thermal environmental characteristics of water bodies in urban areas:

(1) There is a lack of research on the water body cool island effect at the residential area scale and with respect to layout patterns. Most theories concerning the thermal environment regulation effects of water bodies on human living spaces concentrate on macroscopic scales, such as cities and regions [41,42], with a scarcity of research focusing on the direct regulatory effects on residential areas [43].

(2) There is a lack of research on the thermal environment of northern Chinese cities. Residential communities in northern Chinese cities, represented by Tianjin, have unique architectural layout patterns and classifications. Moreover, there are significant differences in the thermal environment regulation requirements of cities in different climatic zones. Research on the impact patterns of the water body cooling effect under summer climate conditions in cold regions of China is insufficient.

This paper explores the impact mechanism of building spaces in residential areas around large planar water bodies with respect to the thermal environment and proposes a method for assessing the cool island effect based on the architectural spatial layout for such residential areas. By investigating the cooling effect of water bodies on residential waterfront areas in Tianjin during the summer and classifying the layout patterns of these areas, this paper validates the impact patterns of various spatial factors in residential waterfront areas with respect to the thermal environment. At the meso level of residential areas, the effective utilization of the water body's cool island effect has been identified. The research findings provide recommendations for waterfront area development planning and for enhancing the quality of residential environments. A flowchart of research ideas is shown in Figure 1.

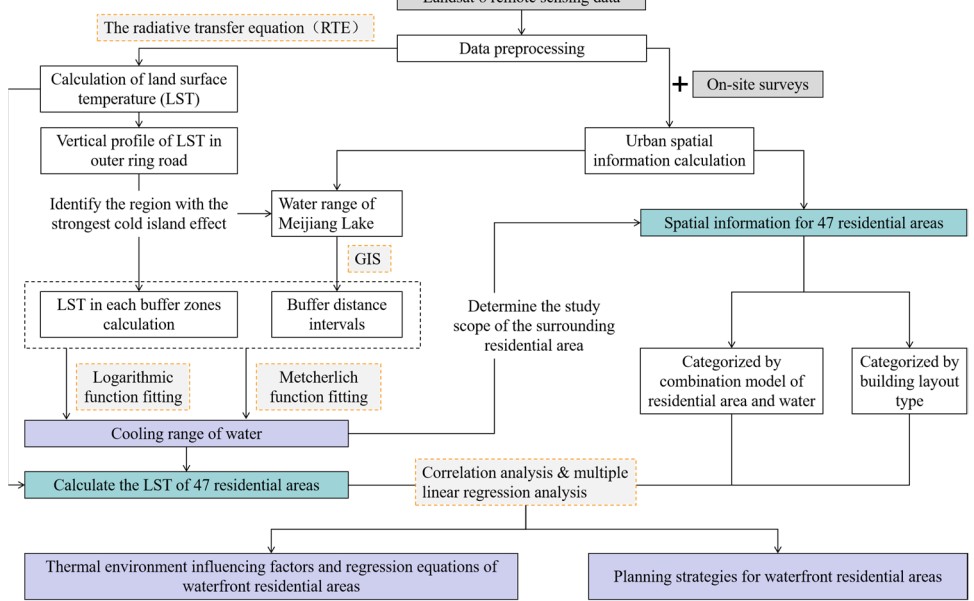

**Figure 1.** A flowchart for the regulating effect of large urban planar water bodies on residential heat islands.

## 2. Materials and Methods

### 2.1. Study Site

Tianjin (38°34′–40°15′ N, 116°43′–118°04′ E), located in the northern part of the North China Plain, serves as the primary research city. Tianjin exhibits typical residential environments, architectural characteristics, and climatic features that are representative of northern Chinese cities. The study area encompasses the urban central district within the outer ring road and the Meijiang Lake region. The selection of this scope is primarily driven by the following considerations: within the outer ring road of Tianjin lies a high-intensity urban development zone that is characterized by concentrated construction, providing the maximum representation of the thermal environmental characteristics of urban built environments. Meijiang Lake, as one of the most significant large-scale blue patches within the outer ring road of Tianjin, serves as a representative example of large planar water bodies within the city. Its surroundings are extensively developed and predominantly residential in function and exhibit diverse residential area layouts, making it an advantageous choice for sample selection.

Based on meteorological data for Tianjin from 2021 to 2022 (NOAA—National Centers for Environmental Information, https://www.ncei.noaa.gov/ (accessed on 12 July 2023)), the monthly average temperature distribution in the city center is illustrated in Figure 2. Typical meteorological days during the summer months from June to August were selected for the study. The research area encompasses Meijiang Lake and its surrounding residential areas, which are situated in the southern part of the urban central district of Tianjin (Figure 3), with a total water surface area of 1.95 square kilometers.

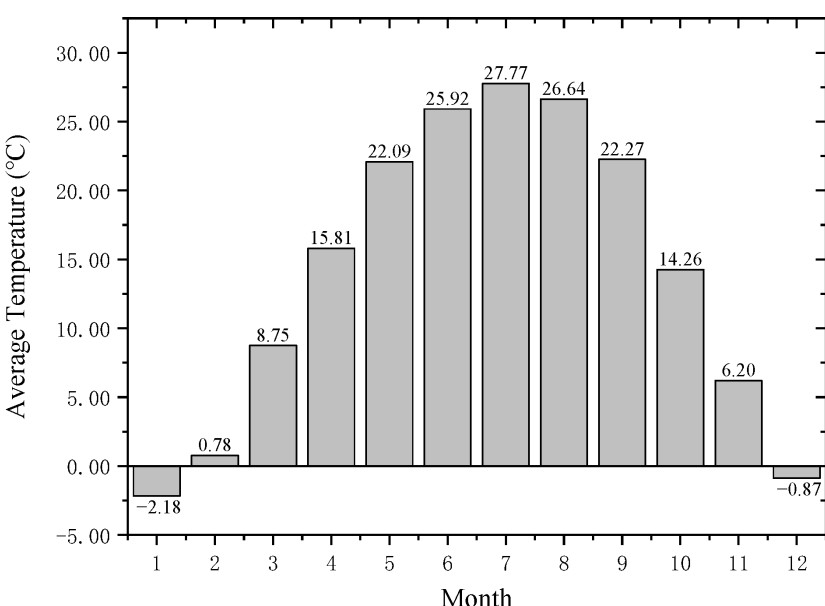

**Figure 2.** Average monthly temperature in the central district of Tianjin from 2012 to 2022.

### 2.2. Data Collection

This study utilized satellite remote sensing imagery from Landsat 8 OLI_TIRS (Operational Land Imager and Thermal Infrared Sensor), provided by the Geographic Spatial Data Cloud Platform (http://www.gscloud.cn (accessed on 15 July 2023)), as the data source for thermal environment analysis. The satellite image selected was acquired on 19 June 2021, with a cloud cover of 0.15%. Surface temperatures were retrieved using the radiative transfer equation (RTE) [44,45] with the assistance of the thermal field variance index from *LST*. Temperature data for the central urban district of Tianjin for the years from 2012 to 2022 were obtained from the National Oceanic and Atmospheric Administration (NOAA) National Centers for Environmental Information (NCEI) website (https://www.ncei.noaa.gov/), and they were obtained specifically from the Tianjin NO. 54527 meteorological station (39°05′ N,

117°03′ E). Water body and building data were sourced from the Tianjin Municipal Bureau of Planning and Natural Resources and supplemented and corrected via on-site surveys.

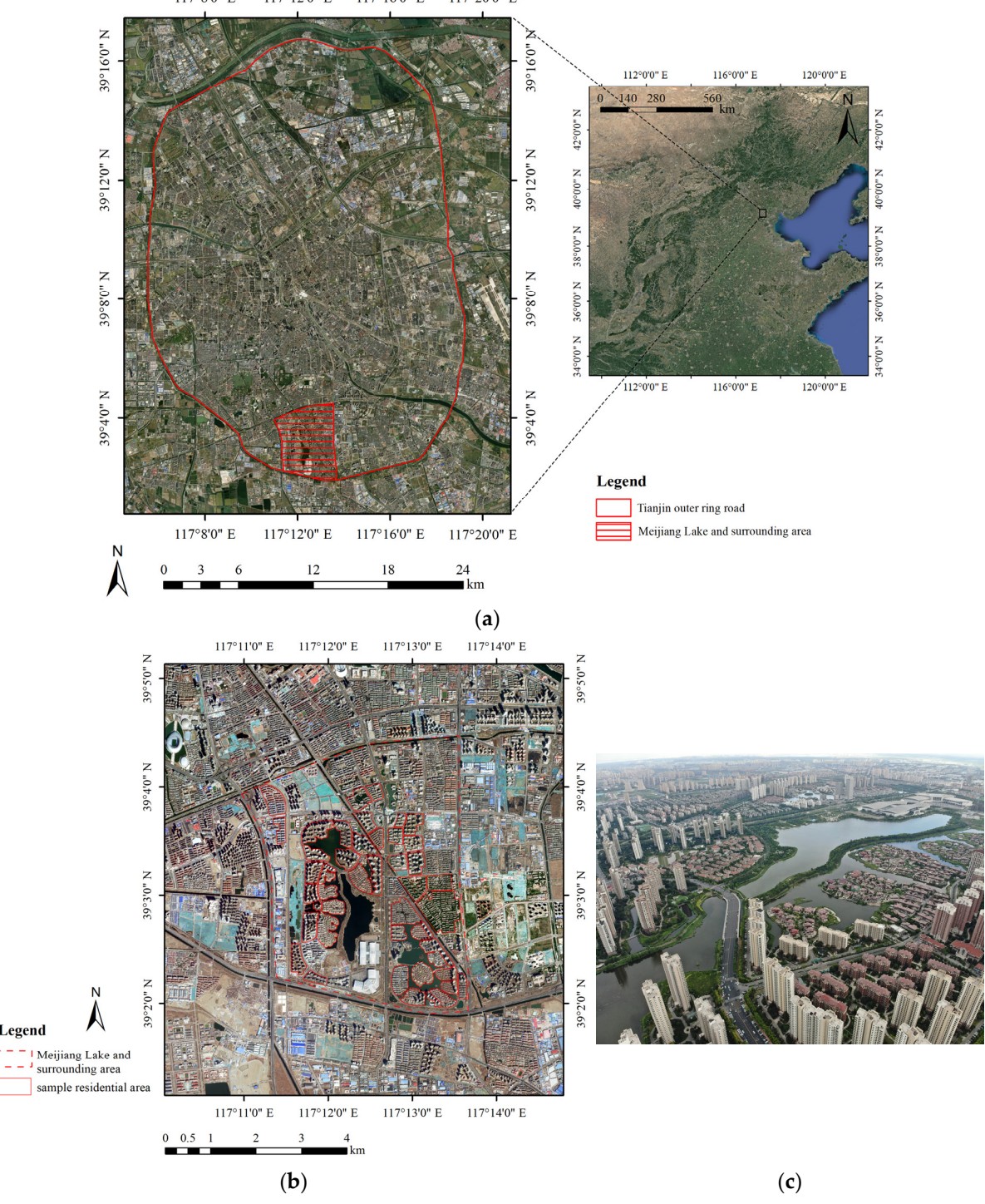

**Figure 3.** Study area showing the following: (**a**) the location of Tianjin's outer ring road and Meijiang Lake and its surrounding areas; (**b**) 47 sample residential areas around Meijiang Lake; (**c**) aerial view photo of Meijiang Lake.

### 2.3. Inversion of Surface Temperature and the Calculation of Water Body Temperature Regulation Range

The surface temperatures within the Tianjin outer ring road area were inverted using the radiative transfer equation (RTE), followed by a profile-based statistical analysis of surface temperatures. This analysis yielded insights into the intensity of the waterbody cool island effect within the urban area of the research scope. The ENVI 5.3 software was employed for this purpose. Initially, the radiometric calibration tool in ENVI 5.3 was used to process the Multi Spectral and Thermal Infrared Spectral 1 of Landsat 8 [46]. It was aimed to convert the image's digital number (*DN*) into radiation intensity value (W·m$^{-2}$·sr$^{-1}$·μm$^{-1}$) via radiation correction Equation (1):

$$Radiance = gain \cdot DN + offset \tag{1}$$

where *Radiance* is the radiation intensity, and *DN* is the digital number for a given pixel. Both the *gain* and the *offset* parameters can be derived from the header file. The subset data via ROIs tool in ENVI 5.3 was utilized to cut the image of each band, using the Tianjin urban area as the boundary. Subsequently, blackbody radiance ($L_T$) was computed using Equation (2) [47]:

$$LT = \left[L_\lambda - L_\mu - \tau(1-\varepsilon)L_d\right]/\tau \cdot \varepsilon \tag{2}$$

where $L_\mu$ denotes upwelling radiance, $L_d$ denotes downwelling radiance, and $\tau$ denotes the transmission through the atmosphere. $L_\mu$, $L_d$, and $\tau$ are available from the Atmospheric Correction Parameter Calculator—NASA (https://atmcorr.gsfc.nasa.gov/ (accessed on 16 July 2023)). $L_\lambda$ denotes the thermal infrared spectral 1 image processed via radiometric calibration. $\varepsilon$ is the surface reflectance, which is calculated via Equation (3) and fractional vegetation coverage [48,49]; then, the value of radiance is converted to the value of surface-leaving radiance:

$$\varepsilon = 0.004PV + 0.986 \tag{3}$$

where *PV* denotes fractional vegetation coverage, obtained by calculating the ENVI 5.3 normalized vegetation index (*NDVI*) tool for the full band image. Finally, Equation (4) is obtained based on the inverse function of Planck's formula, and *LST* is calculated:

$$LST = [k_2/\ln(k_1/L_T) + 1] - 273 \tag{4}$$

where *LST* is the surface temperature (°C). $k_1$ and $k_2$ are calibration constants: $k_1$ = 774.89 W/(m$^2$·sr·μm); $k_2$ = 1321.08 K.

When studying the differences between the urban cold source, the surrounding thermal environment, and their influence ranges, the method of establishing buffer zones is often adopted. The research results of Gudina L. F. et al. [50], Peng J. et al. [51], and Xiao Y. et al. [52] show that buffer zones with a width of 30 m can better reflect the influence range of the cold island and can also more effectively record data such as spatial characteristics and vegetation cover characteristics. Landsat 8 offers varying resolutions across its spectral bands: 100 m for the 10-TIR and 11-TIR bands and between 15 to 30 m for the remaining bands. Therefore, a total of 33 buffer zones were created using ArcGIS 10.8 from the boundary of the water body within the region outwards at 30 m distance intervals (*DI*) (Figure 4).

The water cold island index (*WCI*) was calculated via Equation (5) for different ranges to describe the cooling of the water body relative to the periphery:

$$WCI = LST_{DI\text{-}n} - LST_{water} \tag{5}$$

where $LST_{water}$ (°C) is the average temperature within the water body, and $LST_{DI\text{-}n}$ is the temperature in a buffer zone at *n* meters from the water body's boundary. *WCI* (°C) is defined as the difference between the average *LST* within the buffer zone and the *LST* over the water body. A larger *WCI* value indicates the lesser cooling effect of the water

body. The effective cooling range and maximum effective cooling range were determined via logarithmic curve fitting and the Mitscherlich function. These analyses provided an intuitive depiction of the thermal environmental regulation range of the water body, assisting in the delineation of the residential area scope of the study.

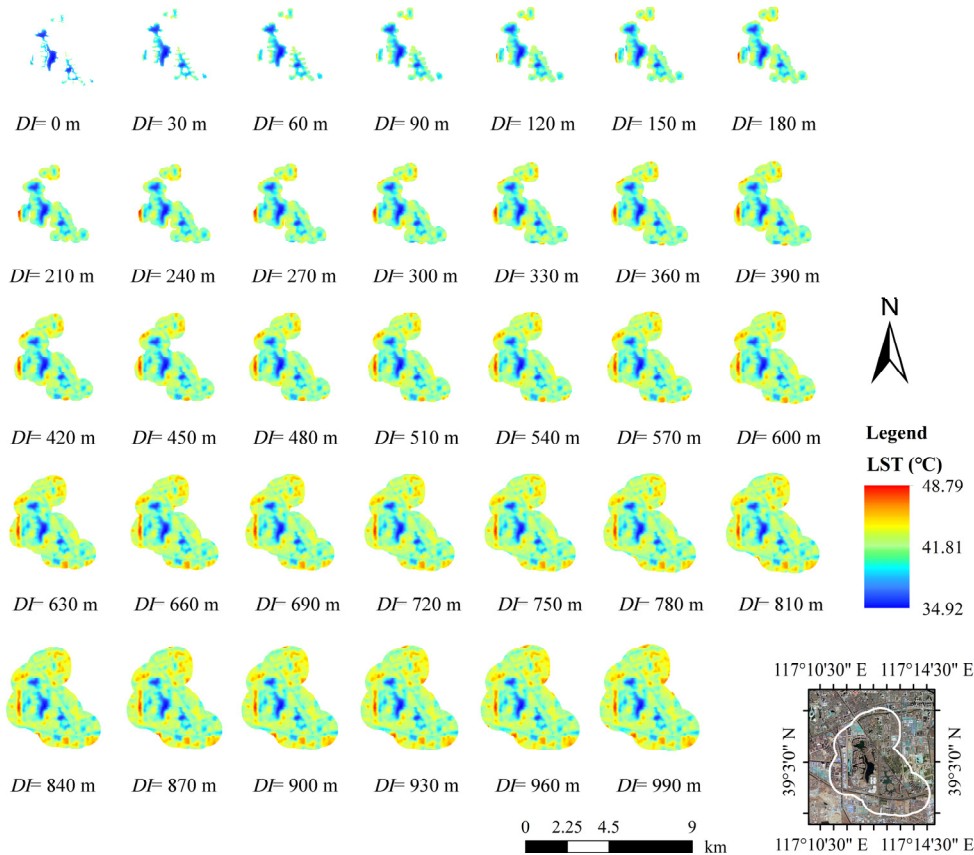

**Figure 4.** *LST* in 30–990 m buffer zones.

## 2.4. Classification of Residential Areas and the Selection of Factors

In order to investigate the thermal environmental regulation role of water bodies on the surrounding residential areas, 47 established residential areas were selected as samples within the effective cooling range of the Meijiang Lake water body. The selection criteria for the samples aim to encompass a broad spectrum of modal combinations associated with water bodies. Additionally, these samples incorporate spatial forms that are prevalent in urban settlement planning across Northern China. This approach ensures that the sample areas are representative, encompassing a diverse range of building heights, widths, and arrangements, thereby enhancing their comprehensiveness and diversity. *LST* data for each of these residential areas were collected and analyzed (as depicted in Figure 5). This analysis aimed to explore the *LST* characteristics of various spatial types of residential areas. By conducting a correlation analysis between spatial features and *LST*, the influencing mechanisms of architectural spatial factors were elucidated.

Based on the layout characteristics of residential areas in northern China, the sampled residential areas were categorized in two ways. This categorization allowed for the investigation of the cooling effects of water bodies on different types of residential areas.

Type I—combination patterns of residential areas and water body (CM): Samples were classified into 4 types based on the spatial relationship between the residential areas and the water body (Figure 6).

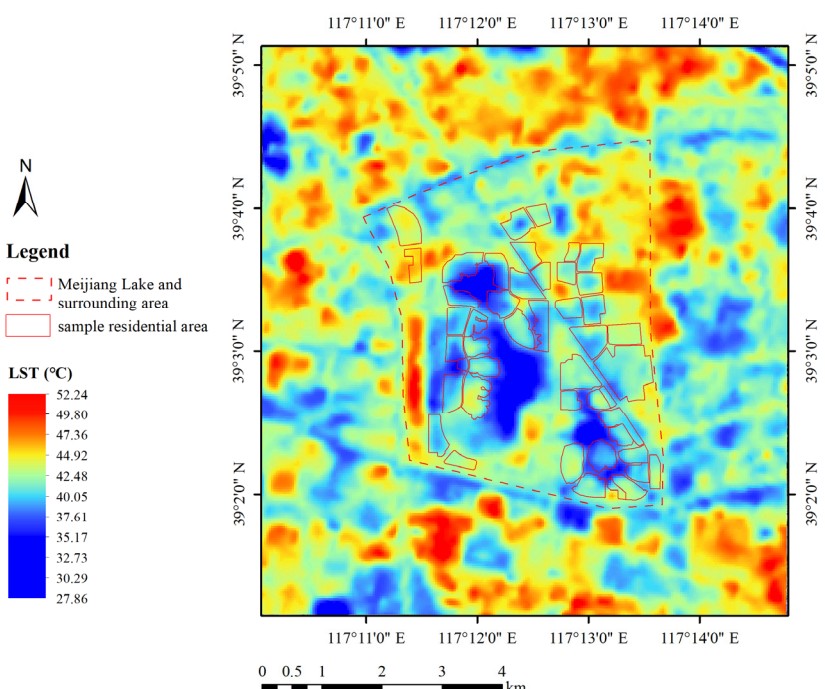

**Figure 5.** *LST* of 47 sample residential areas around the Meijiang Lake.

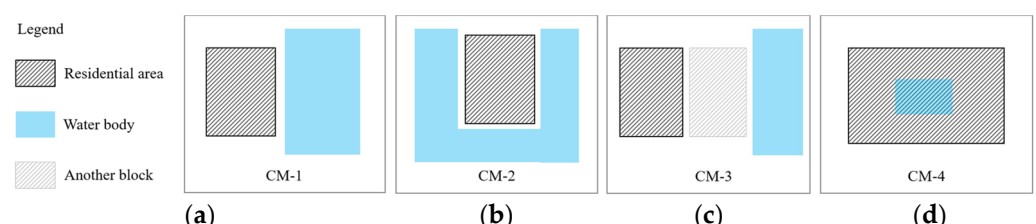

**Figure 6.** Schematic of type I: combination patterns of residential areas and water bodies. (**a**) one side adjacent to water (CM-1); (**b**) surrounded by water (CM-2); (**c**) not directly adjacent to water (CM-3); (**d**) water within the residential area (CM-4).

Type II—layout of the building: In the context of the vast majority of Chinese cities, with Tianjin as a representative case, property developers commonly undertake development and design on a per-residential community basis. Due to constraints imposed by development practices, residential habits, and regulations governing residential area planning, common residential communities can be abstractly categorized into several classic architectural layout patterns. Moreover, based on the requirements outlined in Chinese architectural design regulations concerning elevators and fire safety [53,54], classification can be carried out considering development intensity and building height (Table 1) [55]. This study is conducted at the residential community level, where a correlation analysis between the spatial metrics and thermal environmental characteristics of sampled residential areas enables a clearer understanding of the cooling effects of water bodies on residential areas with different layout patterns and spatial characteristics.

**Table 1.** Tianjin residential area building layout type.

| Layout of Building | Plan Description | 3D Description | Case Selection |
|---|---|---|---|
| Low parallel (villa) (A) | | | |
| Parallel (B) | | | |
| Dispersed (C) | | | |
| Mixed (D) — Mixed-I (D-I) Dispersed + parallel | | | |
| Mixed (D) — Mixed-II (D-II) Parallel + Low parallel | | | |

By conducting correlation analysis and multiple linear regression analysis on the following six spatial metrics in relation to residential areas' thermal environments, we aim to elucidate the impact mechanisms of residential area spatial building factors on thermal environments. This study investigates how residential buildings with different spatial characteristics affect the thermal regulation effects of large water bodies:

1. The distance between residential areas and water (*WD*): The microclimate effect of a water body gradually decreases relative to the area far away from the water body [56]. The length from the nearest point of the bank to the vertical line of the bank is taken as the waterfront distance of the block in this study.

2. Average building height (*H*): The variation in building height affects the roughness variation of an urban area and thus affects the heat island effect. In addition, the vortex and wind shadow regions formed around high-rise buildings will also affect regional ventilation and heat dissipation. The average building height in the block is calculated using Equation (6):

$$H = \left[ \sum_{i=1}^{n} (h_i \cdot A_i) \right] / \sum_{i=1}^{n} A_i \qquad (6)$$

*H*—average building height in the residential area;
$h_i$—the height of each building;
$A_i$—the floor area of each building;
*n*—number of buildings in the residential area.

3. Average building surface width (*L*): The average value of the long side of all buildings in the residential area is one of the metrics that reflect the scale and form of buildings.

4.  Building density (*BD*): Open spaces are crucial factors that influence the microclimate, and the land coverage ratio is employed to describe the proportion of non-building open spaces within a site's total area. This ratio can be substituted with building density as a metric. Higher building densities tend to reduce regional ventilation efficiency, exacerbating the urban heat island effect. Empirically, a building density ranging from 25% to 35% is optimal and recommended for waterfront areas, favoring an approach that is characterized as "sparse in front, dense in back, and well structured" [57]. The calculation method for building density within residential areas is outlined in Equation (7):

$$BD = \left( \sum_{i=1}^{n} A_i \right) / A_T \tag{7}$$

*BD*—building density in the residential area;
$A_i$—the floor area of each building;
$A_T$—total land area of the residential area;
*n*—number of buildings in the residential area.

5.  Floor area ratio (*FAR*): *FAR* reflects the intensity of urban block development. Research has suggested that architectural waterfront clusters with lower *FAR*, coupled with spacious and well-ventilated corridors, tend to exhibit lower temperatures [58]. The influence of *FAR* on microclimatic patterns requires further exploration, and its calculation method is detailed in Equation (8):

$$FAR = \left( \sum_{i=1}^{n} S_i \right) / A_T \tag{8}$$

$S_i$—total building area of the residential area;
$A_T$—total land area of the residential area;
*n*—number of buildings in the residential area.

6.  Normalized difference vegetation index (*NDVI*): *NDVI* is employed to ascertain he vegetation coverage and growth on a land patch. Some studies have demonstrated its significant correlation with urban surface temperatures [59,60]. *NDVI* calculation utilizes the spectral reflectances from Landsat 8 [61], and values in urban range from −1 to 1. Its calculation method is detailed in Equation (9):

$$NDVI = \frac{R_{NIR} - R_{red}}{R_{NIR} + R_{red}} \tag{9}$$

$R_{NIR}$—NIR band (near-infrared band) of Landsat 8;
$R_{red}$—Red band of Landsat 8.

## 3. Results

### 3.1. Regulation Range of Thermal Water Environments in Meijiang Lake

Within the research scope, confined to the area inside the outer ring road of Tianjin city, a vertical cross-section was examined at the center of Meijiang Lake, revealing the distinct formation of a "cold island" phenomenon within this region. The average surface temperature along the vertical profile was recorded as 40.41 °C, with the lowest temperature of 28.37 °C observed over Meijiang Lake's water area (Figure 7).

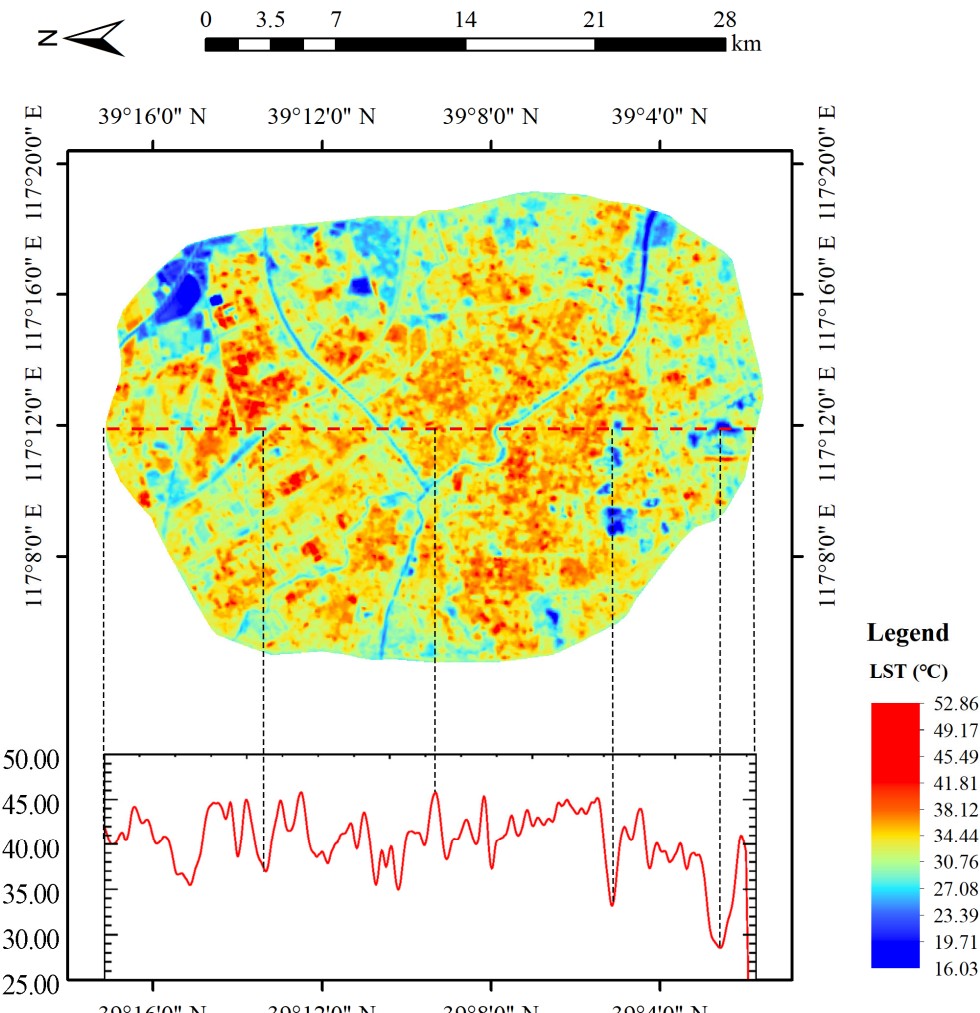

**Figure 7.** Vertical profile analysis of *LST* in the Tianjin outer ring road. The red dashed line is the sectional line. The black dashed lines point from the points on the line to their respective *LST*s.

Buffers were created and *WCI* was calculated as described in Section 2.3. In order to investigate the cooling effect of the Meijiang Lake water body on the surrounding urban area, *WCI* was used as the dependent variable, while *DI* served as the independent variable for function fitting analysis. As observed from the scatter plot, with increasing distances, the average *LST* within the buffer zone exhibited a significant increase, but this increase gradually slowed down until it stabilized. The logarithmic function was fitted to describe their relationship, resulting in Equation (10):

$$y = -3.13042 + 1.30528 \cdot \ln(x), \tag{10}$$

where an adjusted R-squared value of 0.997 is used. The logarithmic function's fit was used to describe the relationship between the two variables, and a range with a function slope greater than 0.01 was identified as the highly effective cooling range [27,62] (Figure 8a). When the tangent slope is 0.01, the *WCI* is 3.24 °C, corresponding to a *DI* of approximately 130 m.

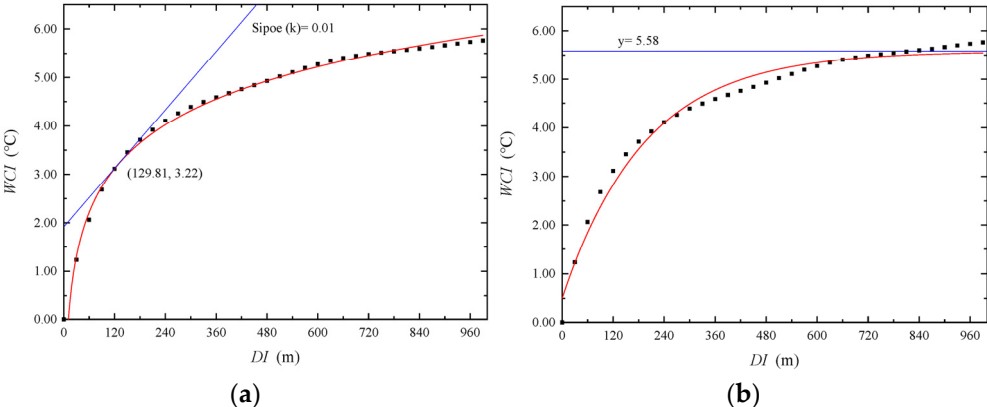

**(a)**  **(b)**

**Figure 8.** Variation characteristics of the water cooling index of different buffer zone distances with respect to Meijiang Lake. Figure (**a**) shows the results of a nonlinear logarithmic function fit, where the red line represents the fitting curve, and the blue tangent line corresponds to the point indicating the temperature change threshold. By examining the change in slope, we can determine that the critical distance at which the cooling effect of the water body significantly decreases is 130 m (129.81 m). Figure (**b**) represents the results of a nonlinear fit using the Mitscherlich function, where the curve of the function gradually converges to the blue function line (y = 5.58). By observing the constant to which the function converges, we can deduce that the critical distance at which the cooling effect of the water body nearly disappears is 810 m.

When *WCI* gradually increased and stabilized, the cooling effect of the water body within this buffer zone range nearly disappeared. This changing pattern aligns with the Mitscherlich function. Therefore, an exponential function was once again employed to fit the relationship between *WCI* and *DI*. The range before reaching the maximum value of the function was considered the effective cooling range. The exponential function was determined as Equation (11):

$$y = 5.5775 - 5.0778 \cdot \exp(-0.00513 \cdot x), \tag{11}$$

with an adjusted R-squared value of 0.982. *WCI* stabilizes when it reaches 5.58 °C, corresponding to a *DI* of approximately 810 m.

In summary, the obviously efficient cooling range of the Meijiang Lake water body extends from 0 to 130 m outwards along the water body's edge, and the effective cooling range spans a distance of 0 to 810 m. Within the range of 130 m to 810 m, there is a certain cooling effect, but the efficiency of the cool island effect significantly decreases. Beyond 810 m, the influence of the cool island effect is nearly negligible. The calculations of *WCI* within 810 m outwards of the water body's edge reveal that the cooling amplitude within the efficient cooling range of 130 m is between 6.05% and 14.44%, with the highest cooling amplitude reaching up to 14.44%.

*3.2. Thermal Environment Characteristics of Residential Waterfront Areas in Different Layout Modes*

The average *LST* values for different CMs were calculated, and they are presented in Figure 9. Notably, CM-3 exhibits a significantly higher average *LST* compared to the other three types, with a difference ranging from 2.85% to 5.36%. CM-1 and CM-2 have similar *LST* values, with CM-2 having the lowest *LST* among the four types. According to one-way analysis of variance results (Table 2) with a $p < 0.05$, post hoc comparisons indicate differences between CM-1 and CM-3, as well as between CM-2 and CM-3, while no differences were observed among the other pairwise comparisons.

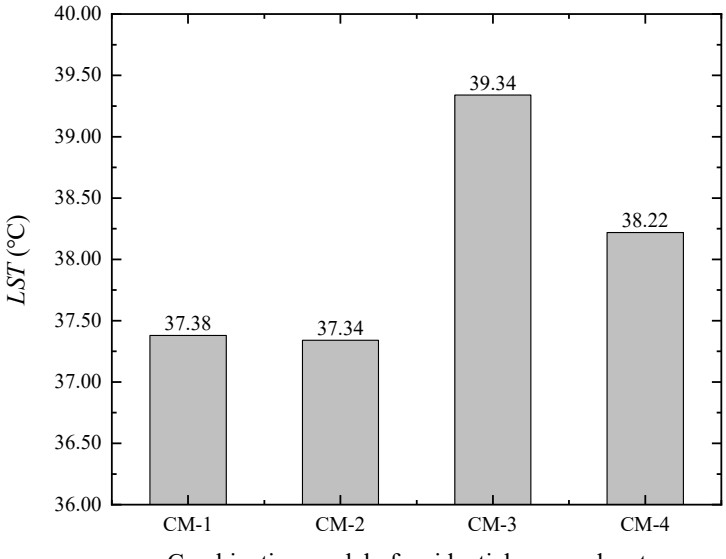

**Figure 9.** Average *LST* of different spatial combinations of residential areas and water.

**Table 2.** Univariate *LST* ANOVA results of different combination models of residential areas and water bodies.

| Group | *LST* (°C) | F [1] | *p* [2] | Post Hoc |
|---|---|---|---|---|
| CM-1 ($n = 17$) | $37.38 \pm 1.33$ | | | |
| CM-2 ($n = 11$) | $37.34 \pm 0.98$ | 8.341 | <0.05 | CM-1 < CM-3 ***[3] |
| CM-3 ($n = 14$) | $39.34 \pm 1.35$ | | | CM-2 < CM-3 *** |
| CM-4 ($n = 15$) | $38.22 \pm 0.51$ | | | |

[1] F is the ratio of the mean square between groups to the mean square within groups, utilized to compute *p*, which determines the significance of differences between groups; [2] *p* represents the level of statistical significance; [3] *** $p < 0.01$.

It is evident that the proximity of residential areas to large contiguous water bodies is closely linked to the thermal environment. As long as residential areas are directly adjacent to large contiguous water bodies, regardless of their specific layout, they tend to achieve relatively effective thermal environment mitigation with minimal differences among them, resulting in an approximately 5.03% lower *LST* compared to areas that are not directly adjacent to water. Moreover, in the design of certain residential areas in China, there is a common practice of incorporating small artificial ponds into community center green spaces and plazas or introducing external natural water bodies to create small ponds, which are aimed at enhancing the quality of outdoor environments and improving aesthetic appeal. While these measures have a modest effect on alleviating the thermal environment, their effectiveness is considerably less pronounced compared to being directly adjacent to large water bodies.

The average *LST* for various building layout types was analyzed (Figure 10), revealing substantial differences in the *LST* among residential areas. Type C exhibited the lowest average surface temperature, while type B displayed significantly higher *LST* values compared to the other types, with differences ranging from 3.87% to 5.58%. Both D-I and D-II featured mixed layout patterns, which are categorized based on different building layouts. Despite variations in layout types between these two classifications, their average surface temperatures were similar. This underscores that the building's layout type is a significant factor influencing *LST*.

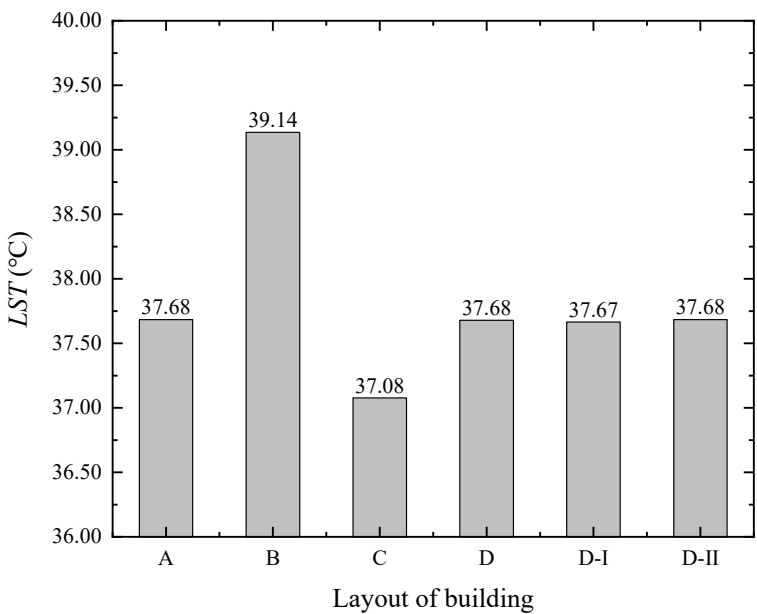

**Figure 10.** Average *LST* of different building layouts.

*3.3. Correlation between Spatial Form and the Thermal Environment of Residential Waterfront Areas*

Compared with *LST*, the change of surface temperature compared with the cold source (water) can reflect the heat island effect of the residential areas more clearly. Heat island intensity represents the temperature difference between two representative measuring points in the heat island effect, often replacing land surface temperature as the dependent variable. Therefore, $\Delta LST$ (the change of surface temperature) (°C) was involved in the Pearson correlation analysis and was used as the dependent variable in regression analysis. It was calculated via Equation (12):

$$\Delta LST = LST_n - LST_{water} \tag{12}$$

where $LST_n$ is the average *LST* of each residential area, and $LST_{water}$ is the average temperature within the water body. Different building layout types fundamentally represent combinations of distinct building heights, building densities, and building widths. Consequently, further investigations will explore the correlations between various parameters. Pearson's correlation analysis was selected to investigate the correlation between $\Delta LST$ and *WD*, *H*, *BD*, *L*, *FAR*, and *NDVI*, across different residential areas to reveal underlying patterns. The results are shown in Table 3.

**Table 3.** Results of correlation analyses between *LST* and *WD*, *H*, *BD*, *L*, *FAR* and *NDVI*.

|  | *WD* | *BD* | *H* | *L* | *FAR* | *NDVI* |
|---|---|---|---|---|---|---|
| *ΔLST* | 0.469 ** | 0.551 ** | −0.410 ** | 0.334 * | −0.120 | −0.401 ** |
| *p* [1] | 0.001 | 0.000 | 0.004 | 0.018 | 0.422 | 0.005 |

[1] *p* represents the level of statistical significance; ** $p < 0.01$; * $p < 0.05$.

$\Delta LST$ exhibited a significant positive correlation with *WD*, *BD*, and *L*. On the other hand, there was a significant negative correlation between $\Delta LST$ and *H*, *NDVI*. $\Delta LST$ showed no correlation with *FAR*. Based on the results of the correlation analysis, it can be observed that *BD* exhibited the strongest correlation with $\Delta LST$, followed closely by *WD*. This suggests that building density and the proximity of residential areas to water bodies are likely the primary factors influencing the thermal environment in residential waterfront areas. Higher building densities and greater distances from water bodies are

associated with poorer thermal environments. Lower vegetation cover also leads to a poorer thermal environment. *BD*, *H*, and *L* are metrics that directly reflect the form of building layouts, and they can also directly reflect the layout types of residential areas (low parallel, parallel, dispersed, and mixed). The characteristics of *LST* under different building layout types are discussed further in Section 4.2 via correlation analysis categorized by building layout types.

Based on the correlation analysis in the previous section, it is evident that Δ*LST* has a significant relationship with *WD*, *BD*, *H*, *L*, and *NDVI*. Therefore, the next step is to delve deeper into understanding the impact of these metrics on Δ*LST*. Since the dependent variable is a continuous numerical variable, a stepwise multiple linear regression analysis is chosen to establish a regression model. Using *WD*, *BD*, *H*, *L*, and *NDVI* as independent variables and Δ*LST* as the dependent variable, a multiple linear regression analysis was conducted (Table 4), resulting in linear regression (Equation (13)):

$$\Delta LST = 6.973 + 0.002 \cdot WD + 8.365 \cdot BD + 0.029 \cdot L - 12.408 \cdot NDVI \tag{13}$$

**Table 4.** Δ*LST* and *WD*, *BD*, *H*, *L*, and *NDVI* regression analysis coefficients, as shown in the table.

| | Mode | Unstandardized Coefficients | | Standardized Coefficients | t | Significance | Collinearity Diagnostics | |
|---|---|---|---|---|---|---|---|---|
| | | B | Standard Error | | | | Tolerance | VIF |
| | (Constant) | 6.973 | 1.278 | | 5.455 | 0.000 | | |
| | *WD* | 0.002 | 0.001 | 0.347 | 3.076 | 0.004 | 0.757 | 1.321 |
| 1 | *BD* | 8.365 | 2.502 | 0.380 | 3.343 | 0.002 | 0.749 | 1.336 |
| | *L* | 0.029 | 0.012 | 0.269 | 2.352 | 0.023 | 0.736 | 1.358 |
| | *NDVI* | −12.408 | 4.474 | −0.322 | −2.774 | 0.008 | 0.717 | 1.395 |

With an adjusted R-squared value of 0.556, the model demonstrates a good fit, suggesting that the regression model fairly identifies the influencing factors of Δ*LST*. Independent variables *WD*, *BD*, *L*, and *NDVI* significantly affect the dependent variable Δ*LST*, accounting for 55.6% of the variance. The order of influence on the dependent variable is as follows: *BD* > *WD* > *NDVI* > *L*, while *H* does not contribute significantly to the variation in the dependent variable.

## 4. Discussion

### 4.1. Factors Affecting the Δ*LST* in Different Combination Models of Residential Areas and Water Bodies

Among the four types of CM, there is a notable difference in the *LST* between areas adjacent to water bodies and those that are not adjacent. This observation indicates that residential areas directly adjacent to water bodies have significantly better environmental conditions. However, even among these water-adjacent residential areas, differences in *LST* exist. To further explore the relationship between the architectural spatial characteristics of water-adjacent residential areas and thermal environment, an analysis of the spatial factors influencing Δ*LST* relative to CM-1 and CM-2 was conducted. Since CM-1 and CM-2 exhibited similar *LST* means and showed no significant differences according to an independent sample *t*-test, they were combined into a single research group referred to as CM-1-2. Similarly, variations in thermal environment were observed among residential areas that are not adjacent to water bodies, prompting a similar analysis of the impact of architectural spatial characteristics on Δ*LST* within this subset. CM-3 and CM-4 were merged into CM-3-4.

A stepwise multiple linear regression analysis was performed for the Δ*LST* of CM-1-2 residential areas, considering *WD*, *BD*, *H*, *L* and *NDVI* as individual variables (Table 5), resulting in the regression shown in Equation (14):

$$\Delta LST_{CM\text{-}1\text{-}2} = 5.474 + 10.293 \cdot BD \tag{14}$$

**Table 5.** Regression coefficients of $\Delta LST_{CM\text{-}1\text{-}2}$ and *WD, BD, H, L*, and *NDVI* regression analysis relative to CM-1-2 residential areas.

| Mode | | Unstandardized Coefficients | | Standardized Coefficients | t | Significance | Collinearity Diagnostics | |
|---|---|---|---|---|---|---|---|---|
| | | B | Standard Error | | | | Tolerance | VIF |
| 2 | (Constant) | 5.474 | 0.507 | | 10.797 | 0.000 | | |
| | *BD* | 10.293 | 2.531 | 0.624 | 4.067 | 0.000 | 1.000 | 1.000 |

The adjusted R-squared value is 0.365. *BD* significantly affects $\Delta LST_{CM\text{-}1\text{-}2}$, while the remaining variables do not exhibit a significant impact. It is evident that in residential areas directly adjacent to large water bodies, the distance from the water body and vegetation coverage are not have a significant effect on the thermal environment. *BD* emerges as the sole factor directly influencing surface temperatures. When *BD* increases by 1%, $\Delta LST_{CM\text{-}1\text{-}2}$ increases by 0.103 °C.

Similarly, for CM-3-4 residential areas, a stepwise multiple linear regression analysis was conducted on $\Delta LST$ in relation to *WD, BD, H, L*, and *NDVI* (Table 6), yielding Equation (15):

$$\Delta LST_{CM\text{-}3\text{-}4} = 14.859 - 28.149 \cdot NDVI \tag{15}$$

**Table 6.** Regression coefficients of $\Delta LST_{CM\text{-}3\text{-}4}$ and *WD, BD, H, L*, and *NDVI* regression analysis relative to CM-3-4 residential areas.

| Mode | | Unstandardized Coefficients | | Standardized Coefficients | t | Significance | Collinearity Diagnostics | |
|---|---|---|---|---|---|---|---|---|
| | | B | Standard Error | | | | Tolerance | VIF |
| 3 | (Constant) | 14.859 | 1.167 | | 12.728 | 0.000 | | |
| | *NDVI* | −28.149 | 5.614 | −0.772 | −5.014 | 0.000 | 1.000 | 1.000 |

The adjusted R-squared value is 0.573, and an increase of 0.01 in *NDVI* resulted in a decrease of 0.281 °C in $\Delta LST_{CM\text{-}3\text{-}4}$. It is evident that the most important factor affecting the thermal environment of residential areas not adjacent to water bodies is the degree of vegetation cover in the residential areas, and the density and width of buildings does not have a decisive impact on $\Delta LST_{CM\text{-}3\text{-}4}$.

### 4.2. Factors Affecting ΔLST in Different Building Layout Categories

The study classified residential areas into three fundamental types: low parallel (villa), parallel, and dispersed, and these are based on the volume, height, and layout of buildings. These types exhibit significant differences in terms of architectural form, residential environment, and spatial experience. Furthermore, they represent the prevalent building layout patterns found in many cities in northern China, using Tianjin as an example. These basic types can also be combined to create various mixed-use residential areas. Analyzing the correlations between $\Delta LST$ and various spatial metrics for different layouts allows for a more purposeful exploration of the influencing factors of thermal environment. It also enables the development of targeted strategies for optimizing the thermal environment by controlling spatial form metrics according to different types of residential areas. It is worth noting that planning regulations for Chinese residential areas often demand different levels of natural lighting for taller building complexes, which typically results in lower building density. The sampled residential areas in this study conform to this pattern, as building density and height exhibit a strong correlation. This correlation also explains why previous chapters found no significant relationship between $\Delta LST$ and *FAR*. *FAR* serves as an indirect metric that is functionally related to building density and height. Within a given residential area, *BD* and *H* are invariably in a direct proportionate relationship with *FAR*. Consequently, *FAR* cannot be effectively used as an influencing factor for $\Delta LST$.

Due to the diverse and irregular nature of D-I and D-II combination patterns, they are not suitable as sample units for discovering patterns. D-I and D-II are, in fact, composed of different combinations of low parallel, parallel, and dispersed building clusters. Analyzing the variations in $\Delta LST$ within these clusters can also achieve research objectives. Hence, the thermal environmental characteristics of D-I and D-II were no longer analyzed separately. Instead, the impact patterns of *WD*, *BD*, *H*, *L*, and *NDVI* on $\Delta LST$ within the three residential area types was investigated—A, B, and C—via stepwise multiple linear regression analysis.

Type A (low parallel): *BD* is the main factor affecting the $\Delta LST$ of type A (Table 7), and regression Equation (16) is obtained:

$$\Delta LST_A = 2.879 + 19.770 \cdot BD \tag{16}$$

**Table 7.** Regression coefficients of $\Delta LST_A$ and *WD*, *BD*, *H*, *L*, and *NDVI* regression analysis relative to type A.

| Mode | | Unstandardized Coefficients | | Standardized Coefficients | t | Significance | Collinearity Diagnostics | |
|---|---|---|---|---|---|---|---|---|
| | | B | Standard Error | | | | Tolerance | VIF |
| 4 | (Constant) | 2.879 | 1.539 | | 1.870 | 0.120 | | |
| | *BD* | 19.770 | 6.181 | 0.820 | 3.198 | 0.024 | 1.000 | 1.000 |

The adjusted R-square is 0.606, which can well explain the variation rule of $\Delta LST_A$. For every 1% increase in *BD*, the average $\Delta LST_A$ of residential areas increases by 0.197 °C.

Type B (parallel): The main influencing factor of $\Delta LST$ in type B is *WD* and *NDVI* (Table 8), and regression Equation (17) is obtained:

$$\Delta LST_B = 14.678 + 0.002 \cdot WD - 30.554 \cdot NDVI \tag{17}$$

**Table 8.** Regression coefficients of $\Delta LST_B$ and *WD*, *BD*, *H*, *L*, and *NDVI* regression analysis relative to type B.

| Mode | | Unstandardized Coefficients | | Standardized Coefficients | t | Significance | Collinearity Diagnostics | |
|---|---|---|---|---|---|---|---|---|
| | | B | Standard Error | | | | Tolerance | VIF |
| | (Constant) | 14.678 | 1.178 | | 12.458 | 0.000 | | |
| 5 | *WD* | 0.002 | 0.001 | 0.351 | 2.491 | 0.026 | 0.992 | 1.008 |
| | *NDVI* | −30.554 | 5.774 | −0.745 | −5.292 | 0.000 | 0.992 | 1.008 |

The adjusted R-square is 0.685, and the degree of interpretation of dependent variables is within the fairly range. For every 0.01 decrease of *NDVI*, the average $\Delta LST_B$ of residential areas increases by 0.306 °C. For every 10 m increase in *WD*, the average $\Delta LST_B$ of residential areas increases by 0.02 °C.

Type C (dispersed): The main influencing factor of $\Delta LST$ in type C was *WD* (Table 9), and regression Equation (18) was obtained:

$$\Delta LST_C = 6.688 + 0.002 \cdot WD \tag{18}$$

The adjusted R-square is 0.258, and the degree of interpretation of dependent variables is within the acceptable range. For every 10 m relative to a large planar water body, the corresponding average $\Delta LST_C$ decreases by 0.02 °C.

**Table 9.** Regression coefficients of $\Delta LST_C$ and *WD*, *BD*, *H*, *L*, and *NDVI* regression analysis relative to type C.

| | Mode | Unstandardized Coefficients | | Standardized Coefficients | t | Significance | Collinearity Diagnostics | |
|---|---|---|---|---|---|---|---|---|
| | | B | Standard Error | | | | Tolerance | VIF |
| 6 | (Constant) | 6.688 | 0.361 | | 18.540 | 0.000 | | |
| | WD | 0.002 | 0.001 | 0.566 | 2.276 | 0.044 | 1.000 | 1.000 |

As observed from the analysis, within residential areas of the same building layout type, neither *H* nor *L* appears to be the primary influencing factor on the thermal environment. Instead, variations in *WD*, *BD* and *NDVI* significantly affect the thermal conditions of these residential areas. Reducing WD effectively diminishes $\Delta LST$ in both parallel (B) and dispersed (C), with negligible differences in their cooling effects. Additionally, in the parallel (B), increasing *NDVI* concurrently further amplifies the reduction in $\Delta LST$. While in the low-parallel (A), *BD* exerts a more pronounced impact on $\Delta LST$. We propose that the predominant classification of low-parallel (A) in the sampled residential areas as CM-1 or CM-2 largely accounts for this phenomenon. This observation further substantiates the notion that residential areas, particularly villas, directly adjacent to water bodies can more effectively decrease $\Delta LST$ by reducing *BD*. Enhancing proximity to water bodies or augmenting green spaces proves less effective in this type of residential area. Implementing specific building density regulations tailored to different building layouts in residential areas can provide a more scientifically sound approach to mitigating the urban heat island effect in waterfront communities.

*4.3. Uncertainties and Limitations*

In this paper, we propose a method to identify spatial indicators that impact the thermal environment in residential areas near planar water bodies and guide residential area planning. However, the data and methods employed have certain uncertainties and limitations. Firstly, the relatively low spatial resolution of Landsat 8 OLI_TIRS data may have marginally affected the results. Notably, many studies on land surface temperature, analyzing the spatial arrangement of buildings [63,64] and different functional blocks [65,66], have utilized Landsat data to draw reliable conclusions. On the other hand, buffers of varying widths introduce uncertainty in outcomes. While Landsat 8 TIR1 offers a 100 m resolution, most other bands provide 30 m resolution. We adopted a 30 m buffer width, commonly used in urban cold island research [50,52]. Peng. J et al.'s research [51] on thermal environment mitigation in parks suggests that the larger the buffer width, the poorer the statistical description of the relationship became. Future research should prioritize the use of higher resolution surface temperature data and focus on urban cold island studies based on this enhanced data.

Selecting independent variables in regression analysis can also introduce uncertainties in the results. The factors influencing *LST* are multifaceted, encompassing not only the indicators discussed herein but also elements such as building materials [67], underlying surface materials [68,69], solar radiation [70], traffic [71] and even population [72]. This research primarily examines the impact of building space on the thermal environment. Consequently, while the variables chosen for this study do not entirely account for the variations in $\Delta LST$, the explanatory power of the regression analysis remains relatively satisfactory. We tried to exclude *NDVI* and solely used *WD*, *BD*, *H*, and *L* for regression analysis. This decision was based on the limited control over urban vegetation coverage and growth compared to other factors. The results have not changed radically, with *BD* and *WD* continuing to be the most significant factors, and the regression model's explanatory power reaching 46.3%. Similarly, we explored incorporating solar radiation-related metrics, such as albedo and global horizontal irradiance (*GHI*) in regression analyses. However, since solar radiation cannot be easily controlled in residential planning, it interfered with our identification of building spatial indicators, so it was not included in independent

variables. Conversely, vegetation coverage, which can be somewhat regulated in residential construction, and the integration of urban water bodies with greening initiatives are significant factors. Therefore, the Normalized Difference Vegetation Index (*NDVI*) was included as an independent variable. While this also reminds us that in addition to the spatial layout and greening of buildings, it is also significant to study the impact of building materials and solar radiation on the thermal environment of waterfront residential areas, which is an important field for future research.

## 5. Conclusions

Leveraging the cooling effect of urban blue and green spaces is currently one of the most effective strategies employed in urban planning to alleviate the urban heat island phenomenon. In particular, large-area water bodies can extend their cooling effects beyond a hundred meters and should be utilized appropriately to enhance the quality of residential areas and living environments. This study draws the following conclusions and provides recommendations for future residential waterfront area planning:

1.  The water surface studied in this paper, located within the Meijiang Lake area, spans an impressive 2 square kilometers. Water bodies of such magnitudes can significantly and directly reduce the temperature of their surrounding areas. The highly efficient cooling range extends up to 130 m from the water's edge, with temperature reductions decreasing from 14.44% to 6.05%, representing the optimal zone for harnessing the cooling benefits of water bodies. It is crucial to give special attention to urban planning and architectural layout within this range, designating it as a priority control zone. Furthermore, the effective cooling range radiating from the water body can extend up to 810 m. Within the 130 to 810 m range, the cooling effect diminishes and gradually stabilizes. It can be designated as a general control zone. Different control standards should be established based on regional control intensity. Efforts should be made to position residential areas as close to the water as possible while simultaneously reducing the building's density and increasing vegetation cover. Diminishing the width of buildings can moderately enhance the thermal environment, although its effectiveness is not as pronounced as the previously mentioned method. These approaches maximize the efficient utilization of cooling effects provided by the water body.

2.  To maximize the effective utilization of the thermal environmental regulation effects of water bodies in residential waterfront areas, it is imperative to ensure that residential areas are in direct adjacency to water bodies and minimize the intervening distance. Notably, once residential areas are already adjacent to the water, whether it is adjacent on one side, two sides, or multiple sides, this specific adjacency pattern does not emerge as the primary determinant that influences the thermal environment. The primary factor affecting the thermal environment in such areas is building density. Residential areas with internal landscaped water features exhibit some cooling effects, yet these are markedly less significant than those experienced by zones adjoining larger bodies of water. Conversely, residential areas that are not directly adjacent to water bodies exhibit the poorest thermal environmental conditions, with an average surface temperature approximately 5.3% higher than that of residential areas that are directly adjacent to water bodies.

3.  Waterfront distance, vegetation coverage, residential area building density and building width are the factors influencing the thermal environment of residential areas. The explanatory power of the independent variables relative to the dependent variable reaches 55.6%. And the most crucial spatial factors are the building density and waterfront distance. Furthermore, in terms of residential area-oriented metrics, although the floor area ratio serves as a commonly utilized metric in China in controlling development intensity, it is an indirect measure derived from building density and building height calculations and does not exhibit a correlation with the thermal environment.

Therefore, when formulating standards pertinent to waterfront areas, building density should be considered as the primary metric for regulating the thermal environment.

4. In light of several prevalent urban layout patterns in northern Chinese cities, distinct control standards should be devised based on their respective characteristics. For low parallel (A) residential areas, which are characterized by relatively lower surface temperatures, effectively regulating building densities represents a viable approach to creating a favorable thermal environment. In the case of parallel (B) residential areas, which typically exhibit the poorest thermal conditions, an average surface temperature that is approximately 5.58% higher than that of dispersed residential areas results. While increasing the vegetation coverage and reducing the waterfront distance, the construction of such residential areas on the waterfront should also be minimized. Conversely, dispersed (C) residential areas, characterized by lower average building densities and higher average building heights, stand to benefit from reducing the distance between the residential area and the waterfront in order to ameliorate the thermal environment effectively.

**Author Contributions:** Project administration, methodology, software, data curation, formal data analysis, writing—original draft preparation, L.W.; investigation, resources, writing-review and editing, G.W.; conceptualization, and supervision, T.C.; validation, and interpretation of results, J.L. All authors have read and agreed to the published version of the manuscript.

**Funding:** This research was funded by National Natural Science Foundation of China (NSFC), Grant agreement ID 52078329.

**Data Availability Statement:** All supporting data are cited in Section 2.

**Acknowledgments:** The authors thank the administration of USGS, EROS and NASA for the free data availability.

**Conflicts of Interest:** The authors declare no conflict of interest.

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
