# Peer review of "The Regulating Effect of Urban Large Planar Water Bodies on Residential Heat Islands: A Case Study of Meijiang Lake in Tianjin"

_land, doi:10.3390/land12122126_

Round 1
Reviewer 1 Report
Comments and Suggestions for Authors
This manuscript explored the regulating effect of Meijiang Lake in Tianjin of China on residential heat island using land surface temperature from remotely-sensed data and multiple linear regression analysis. This work is significant for deep understanding of the urban cool island effect associated with large urban aquatic bodies at a scale of residential area. The result could support the urban planning and enhancing the quality of residential environments locally. However, there are some problems with the writing in this manuscript.
Comments and suggestions are as follows,
1. In the Introduction, authors should simplify the expression of the main line of this study, and then propose the key problem with a progressive structure. Additionally, please add related citations to provide more evidence for these sentences such as Lines 74-77 and line 79.
2. As shown in Figure 4, urban green spaces are distributed in the study area, and also create the cooling island effect locally. So, did the cooling island effect measured by this study take that into account?
3. In the Materials and Methods, the locations of the meteorological stations should be shown to expand the key information for temperature in this study.
4. The land surface temperature is a primary indicator for analyzing the cooling island effect in this study, and then its accuracy should be depicted further.
5. As shown in the equation (1), the land surface temperature was calculated by the thermal infrared sensor 1 with a spatial resolution of 100 meters. The calculated results were then divided into 33 buffer zones at 30 meters distance intervals. Can such a buffer interval effectively capture the spatial details of the land surface temperature changes?
6. In section 2.4, the selection criteria of the 47 samples should be explained further.
7. The material of buildings could be considered into exploring the impacts of buildings on thermal environments.
8. The waterfront distance and building density account for 46.3% of the variance. The expressions of the “effectively” in the line 412 is not appropriate. Changes in the land surface temperature should be adopted as the dependent variable of a regression analysis, and local solar radiation is also suggested to be added to the independent variables.
9. An uncertainty of this study caused by data or methods being used should be further analyzed in the Discussions.
10. Lines 101 and 168, the abbreviations as the CFD and TM should be defined the first time they appear in the main text.
11. The key parameters of Table 2 and Table 3 such as F and P, should be depicted further. It is recommended that Figures 3 and 4 be merged into one figure.
12. The language of this manuscript needs careful editing by a native English speaker.
Comments on the Quality of English LanguageThe language of this manuscript needs careful editing by a native English speaker.
Reviewer 2 Report
Comments and Suggestions for Authors
Dear Authors,
First to point out, that is was a pleasure reading this paper and its clear concept and presentation of facts.
Following are suggestions:
a) In introduction there are places which are not reinforced with references which makes them personal opinion:
Line 56: 'Water bodies, as blue patches within cities, exhibit a cool island effect during the summer due to their lower surface radiation and cooling of the surrounding air' - find and add a reference
Line 62: 'Tianjin, situated in the downstream area of the Haihe River Basin, is where five major tributaries of the Haihe River system converge. It is a city with a relatively high proportion of surface water in the North China region, making the relationship between urban residential environments and surface water environments complex. Tianjin's urban development has been closely
linked to its waterways since its inception, and the relationship between residential areas and water bodies is inseparable' - find and add a reference
Line 68: 'Utilizing water bodies within the city to create favorable microclimate for residential areas and harmonizing the relationship between
water and urban residential environments are key concerns in the planning and development of Tianjin' - find and add a reference
Line 74: 'Efficiently harnessing the cooling effects of large planar water bodies can not only create pleasant outdoor spaces and enhance the quality of living but also reduce heat-related health risks and promote carbon emissions reduction' - find and add a reference
Line 88: 'Currently, most research on the impact mechanism of water bodies on the thermal effects of urban spaces is conducted at macro scales, including studies on urban clusters and city scales.' - find and add a few references to such research.
Line 114: 'Most theories concerning the thermal environment regulation effects of water bodies on human living spaces are concentrated on macroscopic scales such as cities and regions, with a scarcity of research focusing on direct regulatory effects on residential areas' - find and add a few references to such theories
b) Line 186: '... involving procedures such as cropping and radiometric calibration utilizing' - please write EXACT and ALL procedures which are used in preprocessing steps not just 'some'. This is scientific paper and should be specific about subject not just informative.
c) Section 'Planning and recommendation'
Most of text written here is repeated in 'conclusion' - don't do that. I would personally present and conclusions and suggestions in 'Conclusion' section. Either way, do not write same thing in two different sections of paper, mostly when everything is within two pages. Please rewrite and reorganize '4.3. Planning recommendations' and 'Conclusion'
Keep up good work
Regards
